# UCD: Unconditional Discriminator Promotes Nash Equilibrium in GANs

## Abstract

Adversarial training turns out to be the key to one-step generation, especially for Generative Adversarial Network (GAN) and diffusion model distillation. Yet in practice, GAN training hardly converges properly and struggles in mode collapse. In this work, we quantitatively analyze the extent of Nash equilibrium in GAN training, and conclude that *redundant shortcuts by inputting condition in $D$ disables meaningful knowledge extraction*. We thereby propose to employ an unconditional discriminator (UCD), in which $D$ is enforced to extract more comprehensive and robust features with no condition injection. In this way, $D$ is able to leverage better knowledge to supervise $G$, which promotes Nash equilibrium in GAN literature. Theoretical guarantee on compatibility with vanilla GAN theory indicates that UCD can be implemented in a plug-in manner. Extensive experiments confirm the significant performance improvements with high efficiency. For instance, we achieved **1.47 FID** on the ImageNet-64 dataset, surpassing StyleGAN-XL and several state-of-the-art one-step diffusion models. The code will be made publicly available.

## 1 Introduction

Over the past decade, generative modeling has achieved remarkable improvements in various domains, such as data generation (Karras et al., 2020b; Ho et al., 2020; Song et al., 2021b; Tian et al., 2024) and image editing (Shen et al., 2020; Zhu et al., 2022; 2023). It is well recognized that, recent generative models including SD3 (Esser et al., 2024) and GigaGAN (Kang et al., 2023), have demonstrated unprecedented ability of high-quality image generation. Despite the comprehensive categories of generative models, one-step generation is currently the most core task in the literature. Many attempts have been made to explore the potential methodology of one-step generation on large-scale datasets (Sauer et al., 2022; Kang et al., 2023; Song et al., 2023; Song & Dhariwal, 2024; Kim et al., 2024a; Sauer et al., 2024; Lin et al., 2024; Zhu et al., 2025).

Among the aforementioned methods, adversarial training tends to serve as the key to high-quality one-step generation, which originally comes from Generative Adversarial Network (GAN) (Goodfellow et al., 2014). By drawing lessons from Nash equilibrium, GAN employs a generator $G$ and a discriminator $D$, achieving data recovery via a min-max game. Concretely, $G$ endeavors to reproduce samples in data distribution, while $D$ competes with $G$ by distinguishing the synthesized samples. In principle, Nash equilibrium is reached when the training converges. That is to say, $G$ manages to fully recover the data distribution, and $D$ fails to distinguish any further.

In practice, however, Nash equilibrium turns out to be hardly reached in GAN training, in spite of the convincing performance. This leads to poor training stability and the well-known mode collapse issue (Arjovsky & Bottou, 2017; Karras et al., 2020a). Several works have focused on improving GAN training and proposed empirical solutions, yet with only little efficacy or expensive additional computational cost (Yang et al., 2021; Wang et al., 2022; Kang & Park, 2020; Jeong & Shin, 2021; Yang et al., 2022). Therefore, promoting Nash equilibrium is attached great importance to GAN, which not only improves GAN performance but also facilitates other one-step generation methods.

In this work, we first dig into the mathematical foundations of GAN training and propose a novel method to quantitatively evaluate the extent of Nash equilibrium, which is both model-agnostic and loss-agnostic, thus of great robustness. Benefiting from this metric, we argue that *condition injection*

*in $D$ brings redundant shortcuts in backbone thus hinders meaningful knowledge extraction.* To be more detailed, with supernumerary condition signal, $D$ backbone may overemphasize some condition-related features while neglecting others which are more meaningful to adversarial training. In this case, $D$ might be overfitted and sub-optimal. Consequently, when synthesized samples by $G$ are with low conditional likelihood, $D$ frequently fails to extract condition-related features. However, with few features leveraged $D$ cannot supervise $G$ properly, thus mode collapse occurs and Nash equilibrium is hardly realized.

Based on the analyses above, we are devoted to designing an effective methodology to alleviate mode collapse and promote Nash equilibrium. We propose a general solution, namely `UCD`, by leveraging an unconditional discriminator. Our motivation is intuitive – canceling condition signal injection enforces $D$ to extract more comprehensive and robust features. By doing so, $D$ could leverage better knowledge to provide supervision for $G$. We further argue in Theorem 1 that such a methodology is compatible with vanilla theory, and that `UCD` can be implemented efficiently in a plug-in manner. Hence, our work offers a new perspective on improving Nash equilibrium and synthesis performance in GAN literature.

## 2 RELATED WORK

**Generative Adversarial Networks (GANs).** GANs (Goodfellow et al., 2014) are one of the representative paradigms of generative modeling. Formulated as adversarial training between the generator $G$ and the discriminator $D$, GANs manage to recover the data distribution in an implicit modeling manner. Thanks to the rapid improvement on synthesis performance, GANs are introduced to various downstream tasks, including image generation (Karras et al., 2019; 2020b; 2021), image translation (Isola et al., 2017; Zhu et al., 2017; Park et al., 2019; Jiang et al., 2020; Park et al., 2020), and editing (Shen et al., 2020; Shen & Zhou, 2021; Zhu et al., 2022; 2023). Conditional GANs (Mirza & Osindero, 2014) further integrate condition signals, such as class labels (Sauer et al., 2023; 2022), texts (Kang et al., 2023; Zhu et al., 2025), and reference images (Casanova et al., 2021). Such attempts enables expeditious generation and impressive quality of GANs. Despite the aforementioned achievements, GANs usually struggle in large-scale and diverse distributions (*e.g.*, text-to-image and text-to-video generation). To this end, some studies endeavor to improve GANs from different perspectives (Wang et al., 2023; Xia et al., 2024; Zhang et al., 2024; Huang et al., 2024; Xiao et al., 2025). However, currently GANs still seem to be falling from grace on image generation tasks compared with diffusion models (Ho et al., 2020; Song et al., 2021b).

**Improving discriminators of GANs.** Among the literature of facilitating GAN training, many attempts have been made focusing on improving discriminators. Some works explore the functionalities of data augmentation to alleviate overfitting issue, which significantly work under low-data regime (Zhao et al., 2020; Karras et al., 2020a; Jiang et al., 2021). Others take efforts on introduce various extra tasks for the discriminator (Kang & Park, 2020; Jeong & Shin, 2021; Yang et al., 2021; Wang et al., 2022). Although the discriminator could be enhanced to some extent, the huge extra computational cost is not negligible. Yang et al. (2022) proposes to leverage a dynamic discriminator, adjusting the model capacity on-the-fly to align with time-varying classification task. Despite the performance improvements, however, such dynamic training paradigm relies highly on the model structure and might be difficult to apply on complex GANs. Unlike prior works, we focus on rectifying the unilaterally unfair adversarial training and improving GAN equilibrium. It can be implemented in a plug-in manner with almost no additional computational cost.

## 3 METHOD

### 3.1 PRELIMINARY

In the sequel, we only focus on conditional generation with one-hot labels through GANs. Denote by $\mathbf{x}$ the random variable with unknown data distribution $q(\mathbf{x}|c)$ conditioned on $c$. GANs integrate a generator $G$ and a discriminator $D$ with a min-max game, to map random noise $\mathbf{z}$ to sample and discriminate real or synthesized samples, respectively (Goodfellow et al., 2014). Formally, GANs

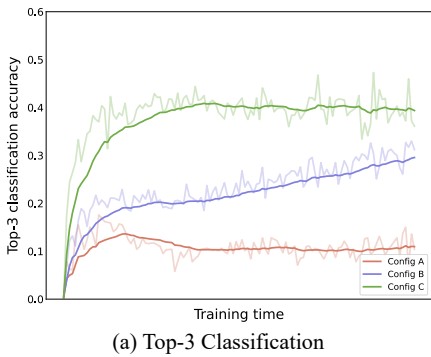 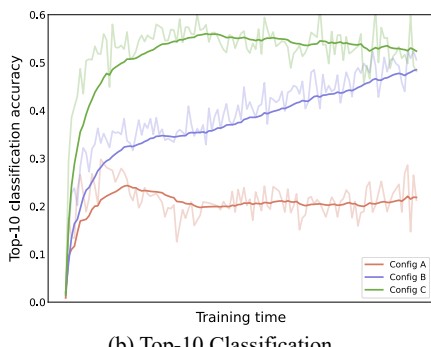

(a) Top-3 Classification  (b) Top-10 Classification

Figure 1: **Visualization of Nash equilibrium** across different models. We integrate $D$ every several iterations into a classification task **without further fine-tuning** according to Eq. (7). A higher classification accuracy suggests better Nash equilibrium. We use Config A (see Sec. 4.1) as the baseline (*i.e.*, the curve in **red**). As a comparison, our UCD is capable of consistently improving Nash equilibrium (*i.e.*, the curve in **blue** and the curve in **green**, respectively). To make a further step, UCD under Config C (Sec. 3.4) achieves more robust $D$, thus better Nash equilibrium. We report both the smoothed values (darker-color curve) and the original values (lighter-color curve) for clearer demonstration, and the horizontal axis suggests the training progress.

attempt to achieve Nash equilibrium via the following losses:

$$\mathcal{L}_G = -\mathbb{E}_{\mathbf{z},c}[\log D(G(\mathbf{z}, c), c)], \tag{1}$$

$$\mathcal{L}_D = -\mathbb{E}_{\mathbf{x},c}[\log D(\mathbf{x}, c)] - \mathbb{E}_{\mathbf{z},c}[\log(1 - D(G(\mathbf{z}, c), c))]. \tag{2}$$

Denote by $p_g(\mathbf{x}|c)$ the underlying distribution from $G$, when the Nash equilibrium is reached, the following equalities hold for any $\mathbf{x}$ and any potential condition $c$:

$$D^*(\mathbf{x}, c) = \frac{q(\mathbf{x}|c)}{q(\mathbf{x}|c) + p_g(\mathbf{x}|c)}, \tag{3}$$

$$p_g(\mathbf{x}|c) = q(\mathbf{x}|c), \tag{4}$$

in which $D^*$ is the optimal discriminator.

### 3.2 Evaluating Extent of Nash Equilibrium

Theoretically, GANs are supposed to achieve Nash equilibrium between $G$ and $D$, so that $G$ can fully recover the data distribution $q(\mathbf{x}|c)$. However, it is hardly achieved in practice and mode collapse frequently occurs. Therefore, evaluating the extent of Nash equilibrium is of great importance to delving into the training dynamics of GAN. Wang et al. (2022) made an empirical and intuitive attempt, which directly visualizes the difference of $D$ logits between real data and synthesized samples. Then larger difference suggests poor Nash equilibrium. However, the difference is significantly affected by the output range and choice of loss functions of $D$, thus fails to clearly and robustly describe disequilibrium.

To this end, we first propose a novel method to better quantitatively describe the extent of the disequilibrium in a model-agnostic and loss-agnostic manner. Assume that for each $\mathbf{x}$ with $q(\mathbf{x}|c) > 0$ and any unrelated condition $c'$, we have $q(\mathbf{x}|c') = 0$. Such an assumption is reasonable for class-conditioned generation, since for unrelated $c'$ the classification probability $q(c'|\mathbf{x})$ must be zero. Then by Bayes' law, we can deduce that $q(\mathbf{x}|c') = q(c'|\mathbf{x})\frac{q(\mathbf{x})}{q(c)} = 0$. Recall that in GAN theory, one first trains $D$ to the current optimality, and then optimizes $G$ accordingly. Therefore when Nash equilibrium is nearly reached, *i.e.*, Eq. (3) holds yet Eq. (4) fails and $p_g(\mathbf{x}|c) \neq q(\mathbf{x}|c)$, for a sample $\mathbf{x}$ with its corresponding condition $c$ and unrelated condition $c'$, we have:

$$D^*(\mathbf{x}, c) = \frac{q(\mathbf{x}|c)}{q(\mathbf{x}|c) + p_g(\mathbf{x}|c)} > 0, \tag{5}$$

$$D^*(\mathbf{x}, c') = \frac{q(\mathbf{x}|c')}{q(\mathbf{x}|c') + p_g(\mathbf{x}|c')} = \frac{0}{0 + p_g(\mathbf{x}|c')} = 0. \tag{6}$$

In other words, $D^*$ is capable of serving as a classifier when Nash equilibrium is almost reached, *i.e.*, for the set $\mathcal{C}$ consisting of all potential conditions, we have:

$$c = \arg\max_{c' \in \mathcal{C}} D^*(\mathbf{x}, c'). \tag{7}$$

On the other hand, poor equilibrium makes $q(\mathbf{x}|c)$ and $p_g(\mathbf{x}|c)$ hardly overlap, leading to $D^*(\mathbf{x}, c') > 0$ thus poor classification accuracy. Pseudo-code is addressed in Algorithm S1.

It is noteworthy that our evaluation concerns only with the classification accuracy, and is independent with intermediate outputs of $D$, thus is both model-agnostic and loss-agnostic. A visualization of training on ImageNet 64 (Deng et al., 2009) is demonstrated in Fig. 1. It clearly demonstrates that vanilla GAN struggles in poor Nash equilibrium, *i.e.*, top-3 and top-10 classifications with the state-of-the-art GAN framework achieve only 10% and 20% accuracy, respectively.

### 3.3 IMPROVING NASH EQUILIBRIUM WITH AN UNCONDITIONAL $D$ (CONFIG B)

Previous works have made attempts to improve GAN equilibrium, yet being too empirical and intuitive (Wang et al., 2022; Yang et al., 2022). In this work, we conclude that redundant shortcuts in $D$ backbone by condition injection disables meaningful knowledge extraction. Concretely, supernumerary condition signal encourages $D$ highly concentrate on condition-related features while neglecting others which are potentially more meaningful to adversarial training. That is to say, $D$ becomes overfitted and sub-optimal. With such a bias, when synthesized samples by $G$ are with low conditional likelihood, $D$ cannot extract sufficiently many effective features to supervise $G$ properly. Therefore, mode collapse will occur and Nash equilibrium is hardly achieved.

In this section, we introduce a novel perspective on improving Nash equilibrium in a neat way, namely UCD. The key motivation is to enforce $D$ to extract more comprehensive and robust features by canceling condition injection. Recall that in vanilla theory of conditional GAN (Mirza & Osindero, 2014), the corresponding condition $c$ is inputted to $D$ simultaneously, which brings shortcuts in $D$ backbone and weakens the robustness of feature extraction. To this end, we propose to train GAN with an unconditional $D$, which utilizes only the data samples with no conditions. In this way, $D$ could extract more versatile feature representations, and then leverages such knowledge to more properly supervise $G$. Formally, denote by $d(\mathbf{x}) \in \mathbb{R}^{\mathrm{card}(\mathcal{C})}$ the classification logits of $\mathbf{x}$, where $\mathrm{card}(\mathcal{C})$ is the cardinality of $\mathcal{C}$. Note that in label-conditioned generation, $\mathrm{card}(\mathcal{C})$ is finite, *e.g.*, 1,000 classes in ImageNet (Deng et al., 2009). Then the overall losses can be written as below:

$$\mathcal{L}_{class} = \mathbb{E}_{\mathbf{x},c}[\mathcal{L}(d(\mathbf{x}), c)] + \mathbb{E}_{\mathbf{z},c}[\mathcal{L}(d(G(\mathbf{z}, c)), c)], \tag{8}$$

$$\mathcal{L}_G = -\mathbb{E}_{\mathbf{z},c}[\log d(G(\mathbf{z}, c))_c], \tag{9}$$

$$\mathcal{L}_D = -\mathbb{E}_{\mathbf{x},c}[\log d(\mathbf{x})_c] - \mathbb{E}_{\mathbf{z},c}[\log(1 - d(G(\mathbf{z}, c))_c)] + \lambda_1 \mathcal{L}_{class}, \tag{10}$$

in which $\mathcal{L}(\cdot, \cdot)$ could be any classification loss, *e.g.*, cross entropy loss or multi-class hinge loss, $d(\cdot)_c$ is the $c$-th component, and $\lambda_1$ is the loss weight. We first prove that Nash equilibrium can be achieved when the training converges, which is summarized as Theorem 1. Proof is in Appendix A.1.

**Theorem 1.** *Let $c$ be the corresponding condition of $\mathbf{x}$, then the $c$-th component of the optimal $d$ training with Eqs. (8) to (10) equals to $D^*(\mathbf{x}, c)$ in Eq. (3). Therefore when training converges we have $p_g(\mathbf{x}|c) = q(\mathbf{x}|c)$, i.e., Nash equilibrium is achieved.*

Theorem 1 claims that our UCD Config B is compatible with vanilla GAN, *i.e.*, the supernumerary classification loss has no influence on the convergence and Nash equilibrium of GAN training. Furthermore, unlike the vanilla GAN pipeline that $D$ is fed with condition signal, the supernumerary classification loss of our UCD enables $D$ to achieve a better backbone for comprehensive feature extraction. This is because the absence of corresponding condition requires more versatile features from $D$ backbone for subsequent classification and adversarial training.

On the other hand, according to Theorem 1, the evaluation method in Sec. 3.2 is still applicable to our UCD. When training with UCD converges and Nash equilibrium is achieved, for each $\mathbf{x}$ and its corresponding $c$, we have $d(\mathbf{x})_c = D^*(\mathbf{x}, c) > 0$. Besides, benefiting from the classification loss, for any unrelated condition $c'$, we can deduce that $d(\mathbf{x})_{c'} < d(\mathbf{x})_c$. Conversely, poor equilibrium suggests poor convergence of classification loss. Therefore, classification accuracy still indicates the extent of Nash equilibrium. Detailed analyses and comparisons are addressed in Sec. 4.3.

Note that although several previous works have come up with similar strategies (Kang & Park, 2020; Jeong & Shin, 2021; Yang et al., 2021; Wang et al., 2022), the consequent huge extra computational cost makes them inapplicable in practice. As a comparison, applying UCD Config B only needs to change the output shape of the final linear head of $D$ from $\mathbb{R}$ to $\mathbb{R}^{\mathrm{card}(\mathcal{C})}$, which is extremely simple and can be implemented in a plug-in manner. Further quantitative evidence of efficacy of UCD Config B is addressed in Sec. 4.3.

### 3.4 Better Nash Equilibrium by Improving Robustness of $D$ (Config C)

It is well known that GAN struggles in training instability and mode collapse issue (Arjovsky & Bottou, 2017; Xia et al., 2024; Xiao et al., 2025), since $D$ is technically a regression-based classifier thus with poor robustness. In other words, $D$ usually appears sub-optimal and $G$ could easily find fake samples that $D$ fails to distinguish. Many attempts focus on improving the robustness of $D$, especially through data augmentation (Zhao et al., 2020; Karras et al., 2020a; Jiang et al., 2021). In this section, we propose to achieve better Nash equilibrium by improving robustness of $D$.

Inspired by the training pipeline of DINO (Caron et al., 2021), the self-supervised learning technique enables robust knowledge distillation through only centering and sharpening of the momentum teacher outputs. To this end, we further integrate a DINO-alike loss on $D$ upon Config B, aiming at more robust feature extraction and more significant supervision for $G$. Therefore, this further facilitates Nash equilibrium. Besides, feeding different views of both real and synthesized samples to $D$ and enforcing consistent feature extraction not only alleviates $D$ from overfitting problem, but also helps to improve the ability of $D$ on the time-varying classification task with continuously changing synthesized samples. Quantitative confirmation is addressed in Sec. 4.3.

Recall that DINO employs a teacher and a student model with an exponential moving average (EMA) technique, yet vanilla GAN leverages only one single $D$. Therefore, for better compatibility we propose to employ $D$ to serve as both teacher and student model with widely used stop-gradient operator. Concretely, two views of one sample are inputted into $D$, in which one is processed with stop-gradient through the same centering and sharpening as DINO while the other is directly used to compute the logit. Detailed implementations are provided in Algorithms S5 and S6. The supernumerary loss $\mathcal{L}_{DINO}$ is added with loss weight $\lambda_2$. Total $D$ loss can be written as below:

$$\mathcal{L}_D = -\mathbb{E}_{\mathbf{x},c}[\log d(\mathbf{x})_c] - \mathbb{E}_{\mathbf{z},c}[\log(1 - d(G(\mathbf{z},c))_c)] + \lambda_1 \mathcal{L}_{class} + \lambda_2 \mathcal{L}_{DINO}. \tag{11}$$

## 4 Experiments

### 4.1 Experimental Setups

**Datasets and baselines.** We apply our UCD on previous seminal GANs on ImageNet 64 dataset (Deng et al., 2009), including R3GAN (Huang et al., 2024) and our self-enhanced R3GAN.

**Evaluation metrics.** We emplot Fréchet Inception Distance (FID) (Heusel et al., 2017) to evaluate the fidelity of the synthesized images, use Improved Precision and Recall (Kynkäänniemi et al., 2019) to measure sample fidelity (Prec.) and diversity (Rec.), respectively. We draw 50,000 samples for each metric evaluation.

**Implementation details.** We use the officially implemented R3GAN[1] (Huang et al., 2024). We implement our self-enhanced R3GAN under Hammer[2] (Shen et al., 2022) as our primary baseline Config A. Concretely, we tune hyper-parameters (*e.g.*, learning rate), change the structure of conditional embedding, add auxiliary classifier head following AC-GAN (Odena et al., 2017), and replace loss function with the LSGAN loss (Mao et al., 2017).

### 4.2 Qualitative and Quantitative Results

We showcase some qualitative results in Fig. 2, demonstrating the efficacy of our UCD. It is noteworthy that by improving the Nash equilibrium with more robust $D$, GAN is capable of synthesizing more diverse samples with better fidelity.

---

[1]https://github.com/brownvc/r3gan
[2]https://github.com/bytedance/Hammer

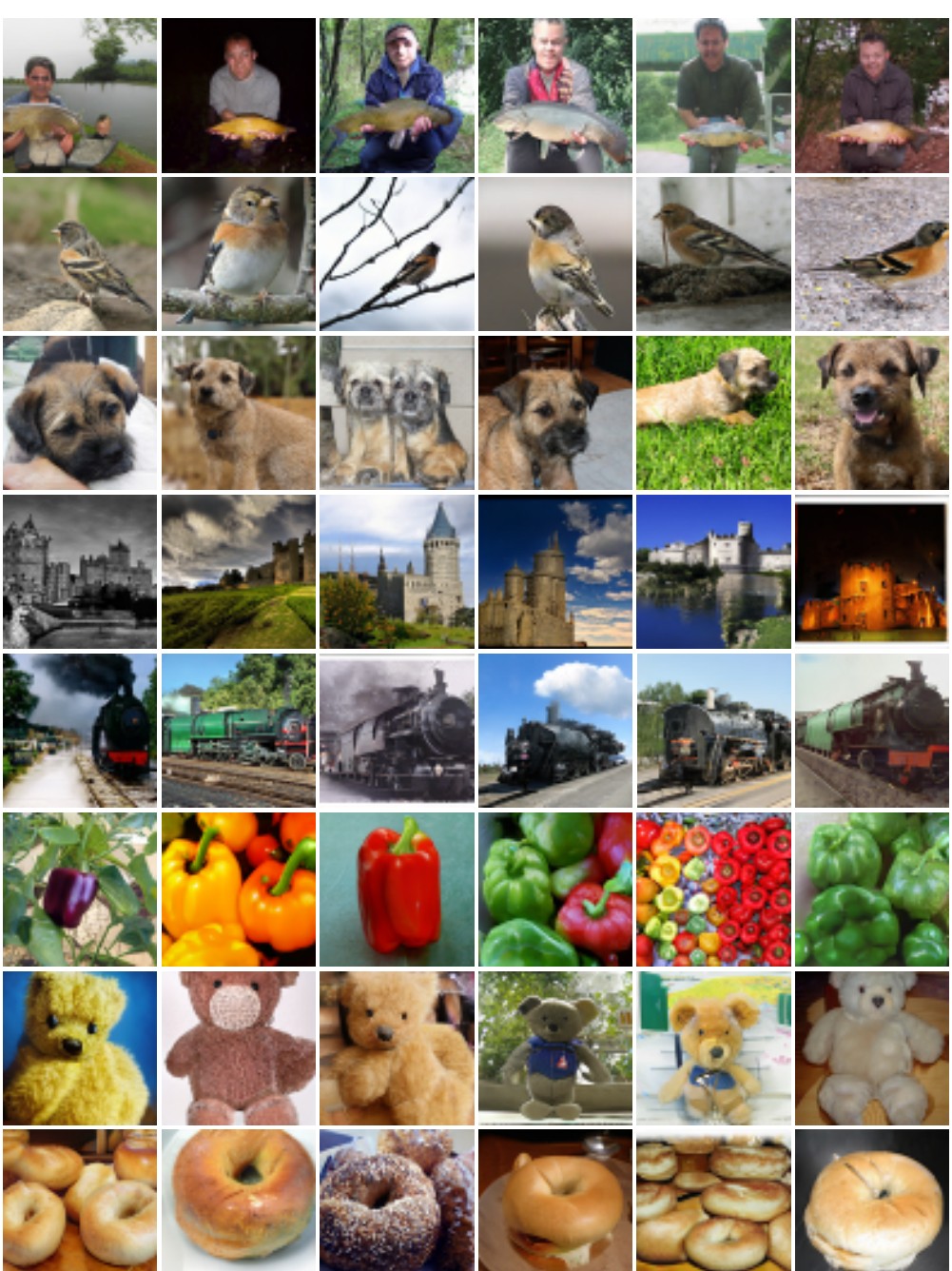

Figure 2: **Diverse results** generated by UCD on ImageNet 64 dataset (Deng et al., 2009). We randomly sample eight global latent codes $\mathbf{z}$ for each label condition $c$, demonstrated in each row.

Besides the exhibited qualitative results, we also report the quantitative results across various state-of-the-art GANs, conveying an overall picture of the capability of our UCD. As is reported in Tab. 1, UCD is able to consistently promote GAN performance by improving the Nash equilibrium. Upon our baseline self-enhanced GAN model (Config A), both Config B and Config C of our UCD significantly improve the FID metric, among which, our Config C achieves 1.47 FID metric and surpasses StyleGAN-XL (Sauer et al., 2022). Note that StyleGAN-XL employs a pre-trained ImageNet classifier to all losses thus potentially leaking ImageNet features into the model, while our UCD utilizes no prior and can be implemented in a plug-in fashion. As for diffusion model methods,

Table 1: **Sample quality** on ImageNet 64x64 (Deng et al., 2009). [†]Methods that utilize distillation techniques. [‡]Methods that utilize auxiliary pre-trained ImageNet classifier to facilitate performance. For clearer demonstration, one-step approaches including GANs and DPMs are highlighted in gray.

| Method | NFE ($\downarrow$) | FID ($\downarrow$) | Prec. ($\uparrow$) | Rec. ($\uparrow$) |
|---|---|---|---|---|
| **ImageNet 64** | | | | |
| PD[†] (Salimans & Ho, 2022) | 2 | 8.95 | 0.63 | **0.65** |
| CD[†] (Song et al., 2023) | 2 | 4.70 | 0.69 | 0.64 |
| PD[†] (Salimans & Ho, 2022) | 1 | 15.39 | 0.59 | 0.62 |
| CD[†] (Song et al., 2023) | 1 | 6.20 | 0.68 | 0.63 |
| DMD[†] (Yin et al., 2024b) | 1 | 2.62 | – | – |
| CTM[†] (Kim et al., 2024a) | 1 | 1.92 | – | – |
| SiD[†] (Zhou et al., 2024) | 1 | 1.52 | 0.74 | 0.63 |
| DMD2[†] (Yin et al., 2024a) | 1 | 1.51 | – | – |
| PaGoDa[†] (Kim et al., 2024b) | 1 | **1.21** | – | 0.63 |
| ADM (Dhariwal & Nichol, 2021) | 250 | 2.07 | 0.74 | 0.63 |
| EDM (Karras et al., 2022) | 79 | 2.44 | 0.71 | **0.67** |
| EDM2-S (Karras et al., 2024) | 63 | 1.58 | **0.76** | 0.60 |
| EDM2-M (Karras et al., 2024) | 63 | 1.43 | 0.75 | 0.62 |
| EDM2-L (Karras et al., 2024) | 63 | **1.33** | 0.75 | 0.62 |
| DDIM (Song et al., 2021a) | 50 | 13.70 | 0.65 | 0.56 |
| DEIS (Zhang & Chen, 2023) | 10 | 6.65 | – | – |
| CT (Song et al., 2023) | 2 | 11.10 | 0.69 | 0.56 |
| iCT-deep (Song & Dhariwal, 2024) | 2 | 2.77 | 0.74 | 0.62 |
| CT (Song et al., 2023) | 1 | 13.00 | 0.71 | 0.47 |
| iCT-deep (Song & Dhariwal, 2024) | 1 | 3.25 | 0.72 | 0.63 |
| StyleGAN2 (Karras et al., 2020b) | 1 | 21.32 | 0.42 | 0.36 |
| StyleGAN2 + SMaRt (Xia et al., 2024) | 1 | 18.31 | 0.45 | 0.39 |
| Aurora (Zhu et al., 2025) | 1 | 8.87 | 0.41 | 0.48 |
| Aurora + SMaRt (Xia et al., 2024) | 1 | 7.11 | 0.42 | 0.49 |
| R3GAN (Huang et al., 2024) | 1 | 2.09 | 0.76 | 0.55 |
| R3GAN + SMaRt (Xia et al., 2024) | 1 | 2.03 | 0.76 | 0.55 |
| R3GAN + MCGAN (Xiao et al., 2025) | 1 | 2.06 | 0.76 | 0.56 |
| R3GAN + UCD | 1 | 1.80 | **0.79** | 0.56 |
| StyleGAN-XL[‡] (Sauer et al., 2022) | 1 | 1.51 | – | – |
| Config A (Ours) | 1 | 1.86 | 0.77 | 0.56 |
| Config B (Ours) | 1 | 1.68 | 0.77 | **0.58** |
| Config C (Ours) | 1 | **1.47** | 0.78 | **0.58** |

we manage to outperform all one-step training methods and loads of the state-of-the-art one-step distillation methods (*e.g.*, SiD (Zhou et al., 2024) and DMD2 (Yin et al., 2024a)).

### 4.3 FURTHER ANALYSES

**Nash equilibrium improvements.** Recall that our UCD is motivated by improving Nash equilibrium by improving robustness of $D$ backbone. In this part, we confirm the efficacy of our method quantitatively via the evaluation method in Sec. 3.2. As demonstrated in Fig. 1, vanilla GAN struggles in poor Nash equilibrium, and achieves only 10% top-3 classification accuracy. As a comparison, Config B enables $D$ to extract more comprehensive and robust features by employing an unconditional $D$, which leads to better equilibrium. To make a further step, Config C additionally integrate DINO-alike loss for better robustness, therefore managing to reach even 40% top-3 classification accuracy and 55% top-10 classification accuracy.

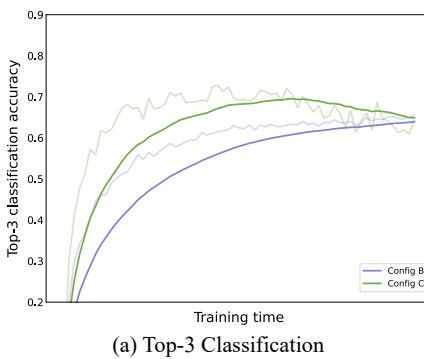 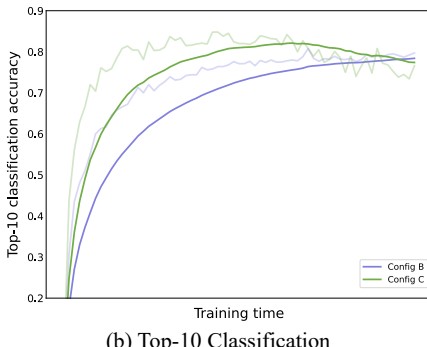

(a) Top-3 Classification          (b) Top-10 Classification

Figure 3: **Comparison** of $D$ backbone between Config B and Config C during training. We **freeze** $D$ **backbone and train a linear classification head** upon it every several iterations. A higher classification accuracy suggests better and more robust $D$ backbone. Compared to Config B (*i.e.* with **blue** curve, DINO-alike loss in Config C with **green** curve enables more robust $D$ backbone. We report both the smoothed values (darker-color curve) and the original values (lighter-color curve) for clearer demonstration, and the horizontal axis suggests the training progress.

Table 2: **Effect comparison** of UCD on ImageNet 64 (Deng et al., 2009). We report the FID metric, the number of parameters, and number of GPUs, respectively.

| | Configuration | FID ($\downarrow$) | $D$ Mparams | # GPUs |
|---|---|---|---|---|
| A | R3GAN (enhanced) baseline | 1.98 | 109.9 | 64 |
| B | + Unconditional $D$ | 1.83 | 114.4 | 64 |
| C | + DINO-alike loss on $D$ | 1.58 | 114.4 | 64 |
| A | R3GAN (enhanced) baseline | 1.86 | 109.9 | 128 |
| B | + Unconditional $D$ | 1.68 | 114.4 | 128 |
| C | + DINO-alike loss on $D$ | 1.47 | 114.4 | 128 |

Table 3: **Ablation study** of hyper-parameters $\lambda_1$ of $\mathcal{L}_{class}$ and $\lambda_2$ of $\mathcal{L}_{DINO}$ in UCD on ImageNet 64 (Deng et al., 2009), respectively.

| | Configuration | FID ($\downarrow$) | $\lambda_1$ | $\lambda_2$ | # GPUs |
|---|---|---|---|---|---|
| A | R3GAN (enhanced) baseline | 2.55 | N/A | N/A | 32 |
| B | + Unconditional $D$ | 2.37 | 0.005 | N/A | 32 |
| | | 2.33 | 0.01 | N/A | 32 |
| | | 2.00 | 0.02 | N/A | 32 |
| | | 3.10 | 0.05 | N/A | 32 |
| C | + DINO-alike loss on $D$ | 1.92 | 0.01 | 0.05 | 32 |
| | | 1.88 | 0.01 | 0.1 | 32 |
| | | 1.89 | 0.01 | 0.2 | 32 |
| | | 1.92 | 0.01 | 0.5 | 32 |

**More robust $D$ backbone.** We further train a classification head upon the frozen $D$ backbone for Config B and Config C, to more clearly demonstrate the robustness improvement of $D$. From Fig. 3 it is clear that integrating DINO-alike loss of Config C enables better classification accuracy, especially at the very beginning of GAN training. This indicates that $D$ backbone appears more robust benefiting from the additional loss, thus capable of leveraging better knowledge. It is also noteworthy that the two accuracies gradually becomes on-par, however, a better $D$ at the early stage could serve as a more meaningful supervision, which is attached great importance to training from scratch. This explains the better Nash equilibrium of Config C from another perspective.

**Effect comparison**. We also report more detailed comparison about the effectiveness of our UCD on ImageNet 64 (Deng et al., 2009), as is demonstrated in Tab. 2. We can first conclude that both Config B and Config C employs the same number of $D$ parameters, appearing close to the Config A baseline. Besides, it is also noteworthy that both Config B and Config C is able to consistently

Table 4: **Comparison** of computational cost on R3GAN (Huang et al., 2024) on ImageNet 64 (Deng et al., 2009). We report the FID metric, maximal GPU memory, average training time for one iteration, and number of GPUs, respectively.

| Method | FID ($\downarrow$) | Max GPU Mem. | Ave. Iter. Time | # GPUs |
|---|---|---|---|---|
| R3GAN (Huang et al., 2024) | 2.09 | 51.40 GB | 2.54 s | 64 |
| R3GAN + SMaRt (Xia et al., 2024) | 2.03 | 52.54 GB | 2.73 s | 64 |
| R3GAN + MCGAN (Xiao et al., 2025) | 2.06 | 51.43 GB | 5.12 s | 64 |
| R3GAN + UCD | 1.80 | 51.50 GB | 2.98 s | 64 |

improve the FID evaluation significantly, in which the two improvements are quite close. That is to say, the effect comparison confirms our motivation of facilitating more robust and comprehensive feature extraction by canceling condition injection in $D$ promotes Nash equilibrium.

**Ablation studies.** We conduct an extensive ablation studies to convey a clear picture of the efficacy of UCD under different hyper-parameters, as reported in Tab. 3. Although we prove in Theorem 1 that the optimality of UCD Config B is consistent with vanilla GAN, too large $\lambda_1$ might weaken the functionality of adversarial training. In other words, $D$ tends to be stuck on classification and fails to distinguish real or fake samples, providing $G$ with less meaningful supervision. It is also noteworthy that the introduction of DINO-alike loss suggest consistent performance improvements, which coincides with the analyses in Sec. 3.4.

**Computational cost comparison.** We report in Tab. 4 the FID performance, maximal GPU memory, average iteration time, and number of used GPUs, respectively. Note that we only change the output shape of the final linear head of $D$ from $\mathbb{R}$ to $\mathbb{R}^{\mathrm{card}(\mathcal{C})}$ with a supernumerary DINO-alike loss, the GPU memory consumption and time cost barely increase. Despite the lazy strategy and omitting the UNet Jacobian of SMaRt (Xia et al., 2024) for better efficiency, we can tell that our UCD slightly increases the training cost but significantly improve the performance.

## 4.4 DISCUSSIONS

It is the shortcuts in $D$ backbone by condition signal injection that hinders Nash equilibrium and restricts the downstream applications, especially leaving GANs lacking further research on conditional generation. Therefore, we hold strong belief that our UCD is attached great importance. Despite the achieved great success on improving Nash equilibrium, our proposed method has several potential limitations. As a supernumerary loss, although we prove in Theorem 1 that it enjoys the same optimality as vanilla GAN, its efficacy depends highly on the choice of the hyper-parameter $\lambda_1$ according to the analyses in Sec. 4.3. Besides, the additional classification loss introduced by our method is designed for label-conditioned generation, and currently incompatible to the mainstream text-to-image or text-to-video generation task. Therefore, how to further conquer this problem (*e.g.*, employing CLIP (Radford et al., 2021) to compute classification loss for text-conditioned generation) will be an interesting avenue for future research, and can improve diffusion model distillation methods. To this end, we hope that UCD could encourage the community to further study the adversarial training and Nash equilibrium in the future.

## 5 CONCLUSION

In this work, we first propose a quantitative method to analyze and evaluate the extent of Nash equilibrium in label-conditioned GAN field. We then theoretically point out a novel perspective to promote Nash equilibrium. By canceling condition injection with an unconditional $D$ and introducing DINO-alike loss to enhance robustness, we propose UCD, a plug-in method which facilitates GAN training and improves synthesis performance. We further provide a proof that the supernumerary loss enjoys the same optimality as vanilla GAN. We conduct comprehensive experiments to demonstrate the significant improvements of synthesis quality on a variety of baseline models, and manage to surpass loads of state-of-the-art diffusion model methods.

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

APPENDIX

# A PROOFS AND DERIVATIVES

In this section, we will prove the theorems stated in the main manuscript.

## A.1 PROOF OF THEOREM 1

*Proof.* Without loss of generality, we use the well-known cross entropy loss for the classification loss $\mathcal{L}(\cdot, \cdot)$, *i.e.*, we have:

$$\mathcal{L}(d(\mathbf{x}), c) = -\log \frac{\exp d(\mathbf{x})_c}{\sum_i \exp d(\mathbf{x})_i}. \tag{S1}$$

Then Eq. (10) can be formulated as below:

$$\mathcal{L}_D = -\int_{\mathbf{x}} q(\mathbf{x}|c) \log d(\mathbf{x})_c + p_g(\mathbf{x}|c) \log(1 - d(\mathbf{x})_c) d\mathbf{x} \tag{S2}$$

$$- \lambda \int_{\mathbf{x}} q(\mathbf{x}|c) \log \frac{\exp d(\mathbf{x})_c}{\sum_i \exp d(\mathbf{x})_i} + p_g(\mathbf{x}|c) \log \frac{\exp d(\mathbf{x})_c}{\sum_i \exp d(\mathbf{x})_i} d\mathbf{x}. \tag{S3}$$

We first compute the $i$-th component of the optimal $d(\mathbf{x})$ for all $i \neq c$. Note that the first integral is independent with each $d(\mathbf{x})_i$, and the second integral can be written as below:

$$- \lambda \int_{\mathbf{x}} (q(\mathbf{x}|c) + p_g(\mathbf{x}|c)) \log \frac{\exp d(\mathbf{x})_c}{\sum_i \exp d(\mathbf{x})_i} d\mathbf{x}. \tag{S4}$$

And for any $\mathbf{x} \in \mathrm{supp}\, q(\mathbf{x}|c) \cup \mathrm{supp}\, p_g(\mathbf{x}|c)$, we can see that $-\log \frac{\exp d(\mathbf{x})_c}{\sum_i \exp d(\mathbf{x})_i}$ reaches its minimum 0 if and only if $\exp d(\mathbf{x})_i = 0$ for $i \neq c$. More generally, for any $\mathbf{x}$ with its corresponding condition $c$, and any classification loss $\mathcal{L}(\cdot, \cdot)$, $\mathcal{L}(d(\mathbf{x}), c)$ can reach its minimum by adjusting $d(\mathbf{x})_i$ to a value independent with $d(\mathbf{x})_c$ for all $i \neq c$, *e.g.*, $\exp d(\mathbf{x})_i = 0$ for cross entropy loss and $d(\mathbf{x})_i = 0$ for multi-class hinge loss. At this time, Eq. (10) is equivalent to the vanilla discriminator loss:

$$\arg\min_{d(\mathbf{x})_c} \mathcal{L}_D = \arg\min_{d(\mathbf{x})_c} \left( -\int_{\mathbf{x}} q(\mathbf{x}|c) \log d(\mathbf{x})_c + p_g(\mathbf{x}|c) \log(1 - d(\mathbf{x})_c) d\mathbf{x} \right). \tag{S5}$$

Then by vanilla GAN theory, we have:

$$d^*(\mathbf{x})_c = \frac{q(\mathbf{x}|c)}{q(\mathbf{x}|c) + p_g(\mathbf{x}|c)}. \tag{S6}$$

Consequently, Eq. (9) also coincides with the vanilla generator loss. According to the native GAN theory, Nash equilibrium is achieved and we have:

$$p_g(\mathbf{x}|c) = q(\mathbf{x}|c). \tag{S7}$$

$\square$

# B PSEUDO-CODES

In this section, we will provide the pseudo-codes of the algorithms proposed in the main manuscript.

## B.1 PSEUDO-CODE TO EVALUATING EXTENT OF NASH EQUILIBRIUM

We below propose the pseudo-codes to evaluate the extent of Nash equilibrium for native GAN and our UCD in Algorithm S1 and Algorithm S2, respectively.

**Algorithm S1** Pseudo-code to evaluate Nash equilibrium of native GAN in a PyTorch-like style.

```python
def run_classification(D, x, c):
    """Defines the function to evaluate Nash equilibrium by classification.

    Args:
        D: Discriminator model.
        x: Data sample.
        c: Label.

    Returns:
        acc: Classification accuracy.
    """
    with torch.no_grad():
        all_c = torch.arange(c.shape[1]).repeat(x.shape[0], 1)
        all_x = x.unsqueeze(0).repeat(c.shape[1], *[1] * x.ndim).transpose(0,1).reshape(-1, *x
    .shape)
        all_logits = D(all_x, all_c).view(-1, c.shape[1])

        pred = all_logits.argmax(dim=1)
        gt = c.argmax(dim=1)
        acc = (pred == gt).sum() / c.shape[0]
```

**Algorithm S2** Pseudo-code to evaluate Nash equilibrium of UCD in a PyTorch-like style.

```python
def run_classification(D, x, c):
    """Defines the function to evaluate Nash equilibrium by classification.

    Args:
        D: Discriminator model.
        x: Data sample.
        c: Label.

    Returns:
        acc: Classification accuracy.
    """
    with torch.no_grad():
        logits = D(x, torch.zeros_like(c))
        pred = logits.argmax(dim=1)
        gt = c.argmax(dim=1)
        acc = (pred == gt).sum() / c.shape[0]

        return acc
```

### B.2    PSEUDO-CODE OF UCD

We below propose the pseudo-codes to implement Config B and Config C of our proposed UCD in Algorithms S3 to S6.

## C    MORE RESULTS

In this part we showcase more qualitative results, as is demonstrated in Figs. S1 and S2. All samples are synthesized with UCD on ImageNet 64 dataset (Deng et al., 2009). It is noteworthy that our proposed UCD manages to improve both the fidelity and the diversity of GAN synthesis, confirming the efficacy of out method.

**Algorithm S3** Pseudo-code to implement Config B of UCD in a PyTorch-like style.

```python
def train_step_config_B(G, D):
    """Defines the function to train one step under Config B.

    Args:
        G: Generator model.
        D: Discriminator model.

    Returns:
        loss: Total GAN loss.
    """
    # First the G loss.
    G.requires_grad_(True)
    D.requires_grad_(False)
    fake_image = G(z, c)
    fake_cls = D(fake_image, torch.zeros_like(c))
    fake_logit = (fake_cls * c).sum(dim=1, keepdim=True)
    g_loss = g_loss_fn(fake_logit)

    # Second the classification loss.
    D.requires_grad_(True)
    G.requires_grad_(False)
    real_cls = D(x, torch.zeros_like(c))
    class_loss = (class_loss_fn(fake_cls, c) + class_loss_fn(real_cls, c)) / 2

    # Finally the D loss.
    real_logit = (real_cls * c).sum(dim=1, keepdim=True)
    d_loss = d_loss_fn(fake_logit) + class_weight * class_loss

    return g_loss + d_loss
```

**Algorithm S4** Pseudo-code to implement Config C of UCD in a PyTorch-like style.

```python
def train_step_config_C(G, D):
    """Defines the function to train one step under Config C.

    Args:
        G: Generator model.
        D: Discriminator model.

    Returns:
        loss: Total GAN loss.
    """
    # First the G loss.
    G.requires_grad_(True)
    D.requires_grad_(False)
    fake_image = G(z, c)
    fake_cls = D(fake_image, torch.zeros_like(c))
    fake_logit = (fake_cls * c).sum(dim=1, keepdim=True)
    g_loss = g_loss_fn(fake_logit)

    # Second the classification loss.
    D.requires_grad_(True)
    G.requires_grad_(False)
    real_cls = D(x, torch.zeros_like(c))
    class_loss = (class_loss_fn(fake_cls, c) + class_loss_fn(real_cls, c)) / 2

    # Third the DINO-alike loss.
    x1, x2 = augment(x), augment(x)
    real_dino_teacher = run_teacher(D, x1)
    real_dino_student = run_student(D, x2)
    real_dino_loss = dino_loss_fn(real_dino_teacher, real_dino_student)
    x1, x2 = augment(fake_image), augment(fake_image)
    fake_dino_teacher = run_teacher(D, x1)
    fake_dino_student = run_student(D, x2)
    fake_dino_loss = dino_loss_fn(fake_dino_teacher, fake_dino_student)
    dino_loss = (real_dino_loss + fake_dino_loss) / 2

    # Finally the D loss.
    real_logit = (real_cls * c).sum(dim=1, keepdim=True)
    d_loss = d_loss_fn(fake_logit) + class_weight * class_loss + dino_weight * dino_loss

    return g_loss + d_loss
```

**Algorithm S5** Pseudo-code to compute teacher outputs in Config C of UCD in a PyTorch-like style.

```python
def run_teacher(D, x, c):
    """Defines the function to forward teacher branch.

    Args:
        D: Discriminator model.
        x: Data sample.
        c: Label.

    Returns:
        dino_teacher: Teacher outputs.
    """
    with torch.no_grad():
        teacher_cls = D(x, torch.zeros_like(c))
        dino_teacher = F.softmax((teacher_cls - center) / temperature, dim=1)

        # Update the center.
        batch_center = dino_teacher.mean(dim=0, keepdim=True)
        center = center * center_momentum + batch_center * (1 - center_momentum)

        # Stop-gradient.
        dino_teacher = dino_teacher.detach()
        return dino_teacher
```

**Algorithm S6** Pseudo-code to compute student outputs in Config C of UCD in a PyTorch-like style.

```python
def run_student(D, x, c):
    """Defines the function to forward student branch.

    Args:
        D: Discriminator model.
        x: Data sample.
        c: Label.

    Returns:
        dino_student: Student outputs.
    """
    student_cls = D(x, torch.zeros_like(c))
    dino_student = F.softmax(student_cls, dim=1)

    return dino_student
```

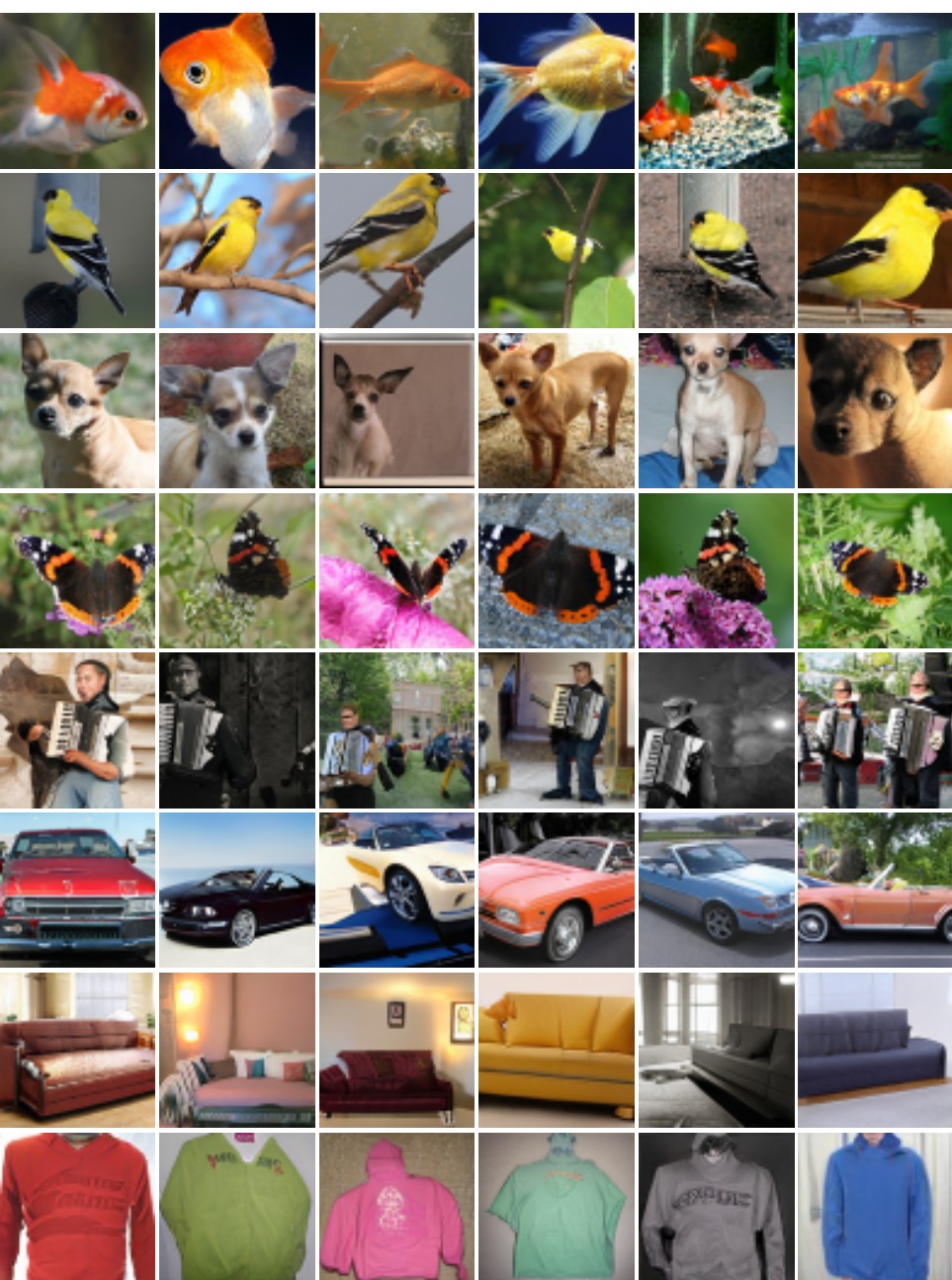

Figure S1: **Diverse results** generated by UCD on ImageNet 64 dataset (Deng et al., 2009). We randomly sample six global latent codes $\mathbf{z}$ for each label condition $c$, demonstrated in each row.

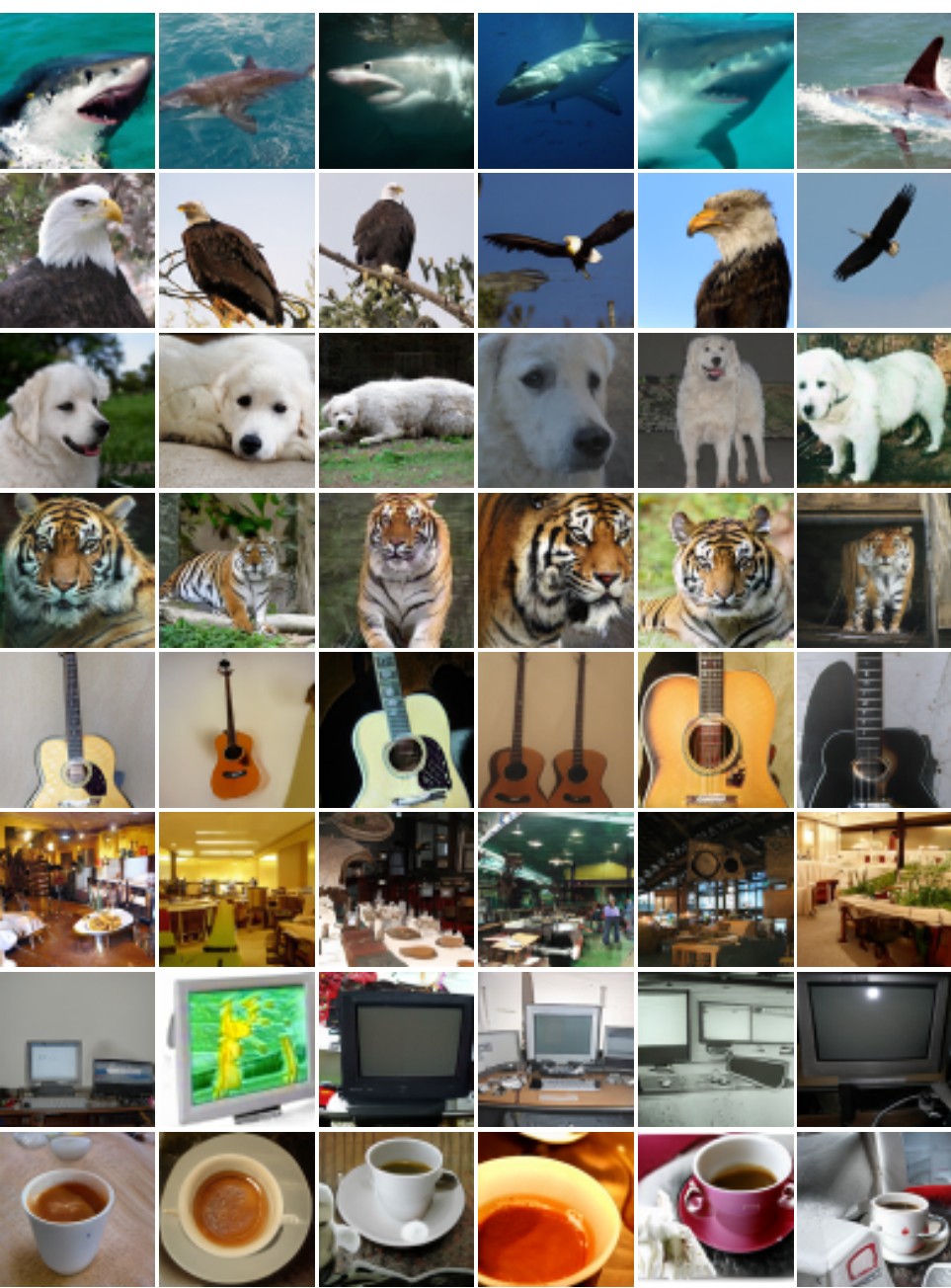

Figure S2: **Diverse results** generated by UCD on ImageNet 64 dataset (Deng et al., 2009). We randomly sample six global latent codes $z$ for each label condition $c$, demonstrated in each row.

