# OpenReview forum: "UCD: Unconditional Discriminator Promotes Nash Equilibrium in GANs"
_ICLR.cc/2026/Conference — ICLR 2026 Conference Withdrawn Submission_

### Official Review · Reviewer_Qgo2 · 2025-10-18

**Soundness:** 2
**Presentation:** 2
**Contribution:** 1
**Rating:** 2
**Confidence:** 4

**Summary:**

The submission proposes to improve convergence in conditional GANs by having the discriminator predict the conditioning-category instead of being fed the category. It is also further suggested to encourage more comprehensive feature learning by multi-tasking a self-supervised DINO loss on the discriminator as well.

Experiments on ImageNet 64 x 64 show competitive performance relative to other GAN variants and diffusion models.

**Strengths:**

* The submission is mostly easy-to-read and follow.

* The method proposed seems to be effective, leading to decent samples.

**Weaknesses:**

* The ideas in the submission are not very original, being quite related to past works such as AC-GAN [1] and Self-Supervised GANs [2], works which have had a lot of follow-up in the literature. In fact, the submission itself seems to use the AC-GAN implementation to add the auxiliary head in their implementation.

* The presentation does not make it very explicit that this is specifically about conditional GANs, which was confusing at times. The fact that only the category-specific head of the discriminator is used makes this entirely a conditional framework, with auxiliary losses.

* The experiments are too small in scale -- image generation has come a long way, and typical comparisons are on much higher resolutions and larger scales.

[1] Conditional Image Synthesis with Auxiliary Classifier GANs, Odena et al., ICML 2017

[2] Self-Supervised GANs via Auxiliary Rotation Loss, Chen et al., CVPR 2019

**Questions:**

No additional questions

---

### Official Review · Reviewer_vdPG · 2025-10-19

**Soundness:** 3
**Presentation:** 2
**Contribution:** 2
**Rating:** 4
**Confidence:** 4

**Summary:**

This article concludes that redundant shortcuts by inputting conditions in D disable meaningful knowledge extraction. Therefore, they propose to employ an unconditional discriminator (UCD), in which D is enforced to extract more comprehensive and robust features with no condition injection. Theoretical guarantee on compatibility with vanilla GAN theory indicates that UCD can be implemented in a plug-in manner. Extensive experiments confirm the significant performance improvements with high efficiency.

**Strengths:**

The article provides a theoretical guarantee that training a GAN with an unconditional discriminator (UCD) alongside an additional classifier can converge to the Nash equilibrium. Additionally, the authors utilize a DINO loss and confirm that it can empirically enhance the diversity of synthesized samples.

**Weaknesses:**

The article presents several issues, which I have outlined below:

1. The theoretical proof has some shortcomings. In Section A.1, how does the author justify that Eq.(s4) equals 0 under all conditions? They only discuss the scenario where \( i \neq c \) and \( d(x)=0 \), but do not provide further analysis for the case where \( i=c \) and \( d(x)=1 \).

2. The DINO loss has not been included in the theoretical proof, making its introduction in the article seem abrupt. Furthermore, the exact formulation of the DINO loss is not provided, which complicates the reader's understanding of its impact on the training of the UCD.

3. The datasets used to validate the effectiveness of the method are limited in scope. The authors should incorporate more datasets with varying resolutions and at least include higher-resolution datasets, as well as a broader range of scene datasets, to verify the generalizability of the UCD.

4. The visualization of the training process is inadequate. The authors should include visualization results that illustrate the effects of the DINO loss, helping readers understand its role in the training of UCD. Additionally, more visualizations from an ablation study addressing how "redundant shortcuts by inputting conditions into D disable meaningful knowledge extraction" should be provided.

5. Due to the aforementioned issues, the theoretical and empirical results do not sufficiently validate the effectiveness of the method.

**Questions:**

1. Please refer to the weakness section.

2. I recommend that the authors expand their analysis regarding the claim that "redundant shortcuts by inputting conditions in D disable meaningful knowledge extraction" from an empirical perspective. Theorem 10 only establishes that an additional discriminator can converge to the Nash equilibrium, but it does not demonstrate that redundant shortcuts impede meaningful knowledge extraction.

3. I suggest that the authors provide more empirical results on the integration of the UCD into a wider variety of GAN models to confirm its plug-and-play capability.

---

### Official Review · Reviewer_arjD · 2025-10-23

**Soundness:** 2
**Presentation:** 3
**Contribution:** 2
**Rating:** 2
**Confidence:** 4

**Summary:**

The paper studies why conditional GANs often fail to reach Nash equilibrium and argues that feeding the condition $c$ into the discriminator $D$ creates “shortcuts” that bias feature extraction and hurt training stability. It proposes a model- and loss-agnostic proxy to quantify proximity to Nash equilibrium by using $D$ as a classifier over conditions and tracking top-k accuracy during training, based on the observation that an optimal $D^*$ should identify the true condition from $x$ when equilibrium is nearly reached, and should reject unrelated conditions (Eq. 7). It introduces UCD, an “unconditional” discriminator that does not receive as input but is trained with an auxiliary classification loss over label logits $d(x)$, with a theorem claiming compatibility with vanilla GAN optimality (Eqs. 8–10; Theorem 1) , and adds a DINO-style self-distillation regularizer on $D$ to improve robustness (Config C; Eq. 11) . On ImageNet-64, the method reports FID 1.47 with Config C, competitive with or better than several one-step baselines, with small overhead relative to R3GAN.

**Strengths:**

Originality. The paper reframes discriminator design and the notion of monitoring equilibrium, beyond prior discriminator-auxiliary tasks that tended to be costly, and is presented as a plug-in change.

Quality: The core claim is backed by a stated theorem. The appendix provides a derivation under common losses. Empirically, UCD improves FID on ImageNet-64.

Clarity: The paper walks through vanilla conditional GAN equilibrium (Eqs. 3–4), defines the equilibrium proxy (Eq. 7) with assumptions stated, and visualizes it (Fig. 1).

Significance. On ImageNet-64, UCD (Config C) reaches FID 1.47 and is competitive with or better than several one-step baselines, even surpassing StyleGAN-XL while not relying on classifier priors.

**Weaknesses:**

1. The proxy assumes that for an “unrelated” class $c'$, $q(x|c')=0$, which is a very strong assumption for natural images with shared structure across classes . More importantly, after adding the classification head in UCD, the metric (top-k accuracy of $D$ over labels) directly aligns with the added loss, so improvements could partly reflect better supervised classification rather than better adversarial equilibrium per se, despite the “model-/loss-agnostic” claim.

2. While the specific “remove $c$ from $D$ + auxiliary classification” combination is neat, the space of discriminator-auxiliary tasks (e.g., AC-GAN-style heads, contrastive/auxiliary tasks) is well explored (the paper itself cites several such lines) . The contribution may be seen as an incremental design choice rather than a fundamentally new objective.

3. Results are on ImageNet-64 only; no evaluation on higher resolutions, other datasets, or non-class labels. The authors acknowledge incompatibility with text-conditioning and gesture at CLIP-based extensions as future work.

4. Ablations show a non-trivial dependence on $\lambda_1$ and $\lambda_2$ (e.g., performance degrades when $\lambda_1$ is too large), suggesting some tuning fragility that should be characterized more fully across seeds and setups.

**Questions:**

1. Given that UCD explicitly optimizes a label-classification head, how do you decouple gains in the top-k equilibrium metric from gains in supervised classification? Could you report an alternative diagnostic independent of $D$'s classification head (e.g., train a frozen, external probe on $D$’s penultimate features or use a separate pretrained classifier for the top-k check)? Please also discuss the practical validity of the assumption $q(x|c')=0$ for unrelated $c'$ and any violations you observe on ImageNet classes.

2. In Appendix A.1, the argument effectively drives non-$c$ components to zero to reduce the classification term and recovers the vanilla objective for the $c$-th component. Does this rely on separability or support assumptions on $q$ and $p_g$? Can you outline conditions under which the joint optimization over $d(\cdot)$ does not interfere with the adversarial signal (e.g., when $\lambda_1$ is finite)?

3. Without feeding $c$ into $D$, how do you ensure that $D$ still provides condition-aware gradients to $G$ (beyond the classification head)? Could you report per-class precision/recall (already partially provided) stratified by class frequency and qualitative failure cases where class alignment goes wrong?

4. Can you include evaluations on at least one additional dataset (e.g., CIFAR-10 or ImageNet-128/256) to test whether UCD’s gains hold beyond ImageNet-64? Also, do Config B/C help unconditional GANs?

5. Please add: (a) sensitivity curves over $\lambda_1$ and $\lambda_2$ with multiple seeds; (b) an ablation that keeps condition injection into $D$ but adds the same classification loss, to isolate the effect of “unconditionalizing” $D$; (c) capacity/architecture ablations for the label head; (d) early-training stability metrics beyond FID (e.g., recall trajectory).

6. You note incompatibility with text-to-image and suggest CLIP-based losses as future work; do you foresee any obstacles to making the equilibrium metric work when conditions are continuous text embeddings rather than discrete labels?

---

### Official Review · Reviewer_AYFe · 2025-10-26

**Soundness:** 1
**Presentation:** 3
**Contribution:** 2
**Rating:** 2
**Confidence:** 5

**Summary:**

The paper introduces UCD (Unconditional Discriminator), a method designed to promote "good" Nash equilibrium in Generative Adversarial Networks (GANs). Its contributions are as follows:

1- Quantitative evaluation of Nash equilibrium – The authors propose a new, model- and loss-agnostic metric that measures the degree of Nash equilibrium during GAN training, based on the discriminator’s classification accuracy across conditions.

2- Identification of conditional shortcut effects – Through analysis, the paper argues that injecting conditioning signals into the discriminator introduces redundant shortcuts that hinder effective feature extraction and equilibrium convergence.

3- Unconditional Discriminator (UCD) framework – The paper proposes removing condition inputs from the discriminator to encourage broader and more robust feature learning, while maintaining compatibility with standard GAN theory. This design can be implemented in a plug-in manner without additional computational cost.

4- Robustness enhancement with a DINO-inspired loss – A DINO-like self-supervised loss is incorporated to further stabilize training and strengthen discriminator feature representations.

5- Empirical validation on ImageNet-64 – Experiments show consistent improvements in image synthesis quality across several GAN baselines, achieving a Fréchet Inception Distance (FID) of 1.47, surpassing prior GAN and one-step diffusion distillation models.

6- Theoretical guarantee – The paper provides a proof that the proposed unconditional discriminator setup preserves the optimality of standard GAN training and converges to the same Nash equilibrium conditions.

**Strengths:**

- Clarity:
While the paper is clearly written overall. There are a few points that should be improved to facilitate reading: Including an expression of Dino's loss and stating the assumptions on which thm 1 implicitly depends.


- Quality of the experimental evaluation:
The authors made a fair amount of experiments to compare with the other sota methods. Their approach yields improved results. The reasons behind these improvements are unclear to me (see weaknesses)

- Originality: I like the idea of introducing simple modifications such as classification loss or Dino's loss, which appears to be effective and original. But I think the paper currently does not do a good job in motivating these modifications (see weaknesses).

**Weaknesses:**

**Soundness 1** The main claim of the paper does not appear to be supported by evidence: "Condition signal encourages D highly concentrate on condition-related features while neglecting others which are potentially more meaningful to adversarial training. That is to say, D becomes overfitted and sub-optimal." Throughout the paper, I see no evidence for this claim, or how the proposed approach addresses this effect. In fact the "unconditional discriminator" appears to be conditional. What's more: any "conditional discriminator" can be expressed in their "unconditional form" as discussed below:

- The authors propose an "unconditional discriminator", yet their discriminator is in fact conditional: it is of the form D(x,c)=  d(x)^{\top}One_hot(c). It just has a particular structure: the conditioning appears by taking the c'th coordinate of the vector d(x). Moreover, since the authors are considering finite/discrete conditions, any discriminator of the form D(x,c) can be represented as a scalar product between some vector d(x) of dim C with a one hot encoding of the class c. Perhaps the details of the discriminator are different, but there is no evidence in the paper explaining how these details are matter: ex: parameter sharing.

 - The results of figure 1 show an improved classification accuracy for the proposed method, however this is not surprising because they are explicitly trained to distinguish between the classes. This does not necessarily mean the learned discriminator is better at discriminating. (section 3.2 suggests that but it relies on the assumption that the classes are well separated, which is unrealistic).



**Soundness of thm 1**: The proof of thm 1 appears to rely implicitly on each class being separated (i.e. each sample x can come only from one class c_0 with p(x|c)=0 for any c\neq c_0.). While this is stated earlier in the text L152, it should appear explicitly in the statement of thm 1. Without such assumption, there is no reason the optimal discriminator takes the for form in eq 3 as stated by the theorem.  Consequently, it is unclear what the objective does when this assumption fails to hold, which is likely to happen in practice. The ablation studies confirm that: larger values of the classification hyper-parameter degrades performance. This should not be an issue, if the loss really preserved the same equilibria as unregularized loss.


**Reasons behind the empirical improvements.** The paper proposes two main modifications to GAN training (with conditional generator): the form of the discriminator and some regulaization losses. The first modification is unconvincing as discussed earlier (in theory equivalent to conditional formulation). The second modification might change the equilibria hence, it is unclear what it does in theory. However, some empirical improvements are shown in the experiments. It remains unclear at this stage to what they are attributed.

I suggest the following ablations/experiments to clarify the effect of each modification:
			- Training a vanilla conditional GAN architecture (vanilla discriminator) using the additional classification loss and/or dino loss: this would show whether the proposed architectural modification really has any effect (I am not convinced it should, but if it does, it is certainly not due to the reasons explained in the paper and more precise explanations should be provided)
			- How generic are the improvements: current experiments are only based on a single experimental setup (Huang et al 2024). But are these observations consistent with other architectural choices, other losses?

**Rethinking the contributions**: It appears to me that the main novelty is to introduce a classification loss (and a dino objective), which is interesting on its own, but the conceptual reasons of these improvements remain unclear to me. I suggest that the authors refocus the paper on explaining why these losses are useful.

**Questions:**

See the weakness section

---

### Note · Authors · 2025-11-12

I have read and agree with the venue's withdrawal policy on behalf of myself and my co-authors.